



# Quantitative imaging of the 3-D distribution of cation adsorption sites in undisturbed soil

Hannes Keck[1,2], Bjarne W. Strobel[2], Jon Petter Gustafsson[1] and John Koestel[1]

[1]Department of Soil and Environment, Swedish University of Agricultural Sciences, P.O. Box 7014, 750 07 Uppsala, Sweden

[2]Department of Plant and Environmental Sciences, University of Copenhagen, Thorvaldsensvej 40, 1871 Frederiksberg C, Denmark

*Correspondence to*: Hannes Keck (hannes.keck@slu.se)

**Abstract.** Several studies have shown that the distribution of cation adsorption sites (CAS) is patchy at a millimeter to centimeter scale. Often, larger concentrations of CAS in biopores or aggregate coatings have been reported in the literature. This heterogeneity has implications on the accessibility of CAS and may influence the performance of soil system models that assume a spatially homogeneous distribution of CAS. In this study, we present a new method to quantify the abundance and 3-D distribution of CAS in undisturbed soil that allows for investigating CAS densities with distance to the soil macropores. We used X-ray imaging with $Ba^{2+}$ as a contrast agent. $Ba^{2+}$ has a high adsorption affinity to CAS and is widely used as an index cation to measure the cation exchange capacity (CEC). Eight soil cores (approx. 10 $cm^3$) were sampled from three locations with contrasting texture and organic matter contents. The CAS of our samples were saturated with $Ba^{2+}$ in the laboratory using $BaCl_2$ (0.3 mol $L^{-1}$). Afterwards, KCl (0.1 mol $L^{-1}$) was used to rinse out $Ba^{2+}$ ions that were not bound to CAS. Before and after this process the samples were scanned using an industrial X-ray scanner $Ba^{2+}$ bound to CAS was then visualized in 3-D by the difference image technique. The resulting difference images were interpreted as depicting the $Ba^{2+}$ bound to CAS only. The X-ray image-derived CEC correlated significantly with results of the commonly used ammonium acetate method to determine CEC in well-mixed samples. The CEC of organic matter rich samples seemed to be systematically overestimated and in the case of the clay rich samples with less organic matter the CEC seemed to be systematically underestimated. The results showed that the distribution of the CAS varied spatially within most of our samples down to a millimeter scale. There was no systematic relation between the location of CAS and the soil macropore structure. We are convinced that the here proposed approach will strongly aid the development of more realistic soil system models.



# 1 Introduction

Soil particle surfaces possess functional groups that are negatively charged. These interact electrostatically with cations and bind them reversibly (cation adsorption sites, CAS). Typically these CAS are positively correlated with the clay and organic matter (OM) content (Murphy, 2015). The amount of CAS per mass of soil is referred to as the cation exchange capacity

(CEC). It is commonly measured in cmol(+) kg$^{-1}$ soil using Ba$^{2+}$ or NH$_4^+$ to replace retained cations. The CEC determines agricultural soil fertility and the nutrient retention capacity (Cardoso et al., 2013; Murphy, 2015) Furthermore, it influences heavy metal retention (Bhattacharyya and Gupta, 2008; Gomes et al., 2001) and the mobility of cationic pollutants in soils (Figueroa-Diva et al., 2010; Gevao et al., 2000; Porfiri et al., 2015).

The content and quality of clay or OM varies within soils depending on the bedrock, soil type, pedologic history, land use

practices, vegetation cover, climate factors and biological activity (Guo and Gifford, 2002; Horn, 1987; Leue et al., 2010; Teferi et al., 2016). Illuvial clay mineral accumulation in argic horizons on aggregate surfaces and macropore walls and the formation of clay cutans leads to a spatially heterogeneous distribution of clay (FAO, 2014). (Horn, 1987) attributed a pronounced increase in fine material and CEC at aggregate surfaces to the aggregates shrinking and swelling activities, leading to separation of fine and coarse material. Biopores (e.g. earthworm borrows, root channels) often create preferential

flow paths and hot-spots of biological activity with higher OM contents compared to the surrounding soil matrix, leading to a spatially heterogeneous distribution of OM and CAS often in relation to the location of macropores (Bundt et al., 2001a, 2001b; Kögel-Knabner et al., 2008; Nielsen et al., 2015). Thus, spatial variation of clay minerals and OM content in the soil profile affects the spatial distribution of CAS (Bundt et al., 2001b; Ellerbrock and Gerke, 2004; Horn, 1987). A higher density of CAS along macropores can be expected, which originates from biological activity (biopores) or is influenced by

clay accumulation. The effect of a higher or lower CEC, especially within the macropore sheaths compared to the soil matrix on soil chemical transport processes, is little investigated and still debated in literature (Jarvis, 2007). Considering a scenario in which solute transport bypasses 90 % of the bulk soil volume, as found by (Koestel and Larsbo, 2014), the location of CAS will influence the performance of mechanistic soil system models, especially when modeling short distance solute transport.

3-D X-ray imaging techniques bear great potential to study and illustrate relationships and feedback mechanisms between structural, hydrological and biochemical soil properties within undisturbed soil (Peth et al., 2014; Thieme et al., 2003; Tracy et al., 2010). Soil pore structure provides a spatial boundary that determines the accessibility of water, air and nutrients as well as separating biological processes within soils by compartmentalization (Ruamps et al., 2011; Young et al., 2008). These influences of 3-D soil structure on local processes can be studies using 3-D X-ray imaging. Larsbo et al. (2014) used 3-D X-

ray scanning to investigate the influence of macropore network characteristics on preferential solute transport. They found that soils with a well connected and large macroporosity decrease preferential solute transport due to increased diffusion and





exchange with the soil matrix. Naveed et al. (2016) investigated the soil structure by X-ray imaging to test prediction models of biopore and matrix dominated water and air flows. Hapca et al. (2015) used a statistical approach to predict 3-D distribution of several elements in undisturbed soil by combining 3-D X-ray images and 2-D SEM-EDX data. Mairhofer et al. (2016) visualized plant root interactions in undisturbed soil and Ahmed et al. (2016) visualized interactions of plant roots,
phosphate fertilizers and soil structure over time in 3-D using X-ray imaging.

A common method in X-ray computed tomography is to visualize the objects or processes of interest by enhancing their X-ray photon attenuation using contrast agents. Peth et al. (2014) imaged the OM distribution in soil using osmium as a contrast agent, which is known to bind strongly to OM. Roscoat et al. (2014) proposed a method to visualize biofilms in porous media using chloronaphtalene as a contrast agent. Koestel and Larsbo (2014) used iodide to increase the X-ray
photon attenuation of water when studying water flow in an undisturbed soil column. A suitable contrast agent should contain element(s) with a higher atomic number than other common elements present in the soil. Furthermore, it must have an affinity to bind to or to dissolve in the object of interest (Van Loo et al., 2014). Such an agent increases the electron density and the X-ray photon attenuation locally by absorbing or scattering a proportion of the primary X-ray beam (Wildenschild et al., 2002). $Ba^{2+}$ is a suitable contrast agent in X-ray computed tomography of soils (Van Loo et al., 2014). It
has a relatively high electron density (54) and a high affinity to replace other cations and bind to CAS. Furthermore, it is present in natural soils in rather low amounts and used in standard methods to measure the CEC. Therefore, it is a prime candidate for labeling CAS for 3-D X-ray imaging in undisturbed soil.

To our knowledge spatial variations in the CEC in undisturbed soil cores were not yet imaged in 3-D. Therefore, the objectives of this project are (i) to visualize the CAS of undisturbed soil samples in 3-D using an industrial X-ray scanner
and $Ba^{2+}$ as a contrast agent, (ii) investigate the spatial distribution of CAS with respect to the distance of soil macropores and (iii) to relate the CEC as determined by 3-D image analyses of the $Ba^{2+}$ saturated soil sample with the CEC determined by a common laboratory analysis using ammonium acetate.

## 2. Material and methods

### 2.1 Site description and sampling

Eight undisturbed, natural soil samples were taken at the end of November 2015 with aluminum columns (height: 5 cm, diameter: 2.2 cm) such that approximately half of the column volumes were filled with soil. Samples numbers (SNO) 1–4 were taken from an agricultural long term field-trial, the Swedish soil fertility experiments (Kungsängen, R3-9001; Table 1; Kirchmann, 1991). The Kungsängen site was established in 1963 near Uppsala and the soil was classified as a gleyic cambisol (Holmqvist et al., 2003). SNO1 and SNO3 were sampled from a depth of 3–5 cm and SNO2 and SNO4 from a
depth of 35–38 cm corresponding to the plow pan after tillage. These samples had a high clay content (Table 1) and showed



earthworm activity. SNO5–7 originated from a soil located in a marshy depression which was periodically water logged. This soil is high in organic matter (OM) and classified as heavy clay (Table 1 and Table 2). SNO8 originates from an organic matter-rich soil developed in loamy sand under a pine forest (Table 1 and Table 2). In addition, two artificial samples of fine sand and several clay and peat aggregates were included in the experiments (SNO9 and SNO10) to visualize the difference in

$Ba^{2+}$ binding efficiency of clay and organic matter separately.

## 2.2 Laboratory analyses

To evaluate the precision of the $CEC_{Ba}^{2+}$ determined through the method described below, the $CEC_{NH4}^{+}$ was determined by the ammonium acetate method. The samples were sieved (< 2 mm) and air dried and the exchange sites were saturated with ammonium ions at a buffered pH of 7. The $CEC_{NH4}^{+}$ was determined by replacement of $NH_4^+$ by $K^+$ and measuring the $NH_4^+$

contents with a Tecator flow injection analyzer (Foss A/S, Denmark). The soil pH was determined in deionized water for all samples using a PHM 93 pH meter equipped with a Radiometer combination electrode (Radiometer A/S, Copenhagen, Denmark). The particle size distribution was analyzed by sedimentation after removal of carbonates and organic matter by using hydrochloric acid (1 mol $L^{-1}$) and hydrogen peroxide (30 %), respectively. The soil texture classes were determined according to FAO (2006). The total carbon contents were analyzed by the loss of ignition method according to SS-ISO

13878 for each sample location (TruMac CNS, LECO Corporation, MI, USA). The bulk density was obtained after all analyses were completed and all samples dried at 105 °C by gravimetry. The soil mass was then related to the sample volume obtained by the 3-D images. The bulk densities were corrected for the calculated $Ba^{2+}$ mass adsorbed by each sample.

## 2.3 X-ray computed tomography imaging

The GE Phoenix v|tome|x m X-ray scanner installed at the Department of Soil and Environment at the Swedish University of

Agricultural Sciences, Uppsala was used in this study. It is equipped with a 16 inch monitor (GE DRX250RT) and a 240 kV X-ray tube with a tungsten target. The samples where scanned at a maximum photon energy of 80 kV and an electron flow of 250 µA. The 3-D images where obtained by combining 2000 radiographs taken over a time of 46–90 minutes, depending on the density of the sample, thus the exposure time per radiograph was 333– 1000 µs. The radiographs were inverted using the GE software datos|x (version 2.1) and exported as 16 bit 3-D Tiff images with a voxel size of 20 µm.

## 2.4 Experimental setup

First we acquired an 3-D X-ray contrast image of the KCl (0.1 mol $L^{-1}$) and the $BaCl_2$ (0.3 mol $L^{-1}$) solutions in separate plastic vials and the aluminum wall of the soils columns was obtained and used to sample the corresponding gray values (Fig. 1). These were used to relate gray values to densities and ultimately to $Ba^{2+}$ mass.



The soil samples were first placed on sand beds in plastic cups and saturated in a desiccator under a near-vacuum with a de-gassed KCl solution (0.1 mol L$^{-1}$) to avoid air entrapment inside the columns. The samples were slowly saturated from below. Residual ions in the soil columns were washed out by daily removal of the supernatant and replenishment of the KCl solution outside the column (Fig. 2). Furthermore, the soil was given time for swelling. The electrical conductivity (EC) of

the supernatant was measured at regular intervals (Device: Cond 3310, WTW GmbH, Weilheim, Germany) and the treatment stopped after the EC in the supernatant had reached the EC of the KCl solution with a deviation of max. 2.5 %. Each sample was scanned with the X-ray scanner in 3-D resolution to obtain a reference images for later processing steps. No air entrapment was found in the reference images upon visual inspection. Figure 3 illustrates the sequence of the individual steps undertaken to conduct the experiment.

All samples were then carefully transferred into new plastic cups filled with the 0.3 mol L$^{-1}$ BaCl$_2$ solution. The samples were slowly saturated with BaCl$_2$ from the bottom up. In the following, the supernatant was daily removed and the BaCl$_2$ solution outside the column was replenished (Fig. 2). In this fashion the resident KCl solution was flushed out and cations on the CAS were exchanged with Ba$^{2+}$. During the Ba$^{2+}$ saturation process the ECs of the supernatants were measured and X-ray images taken at regular intervals in order to find the time of Cl$^{-}$ breakthrough and to monitor the spatial distribution of Ba$^{2+}$

within the samples. The treatment was stopped after the EC in the supernatant of each sample had reached the EC of the BaCl$_2$ solution (max. tolerance of 2.5 %) and the gray values in the 3-D X-ray images showed a temporally stable distribution. This was the case after 25 days and removal of 160 mL cumulative supernatant, which corresponds to an average of 15 times the soil columns volume.

To ensure that all non-adsorbed barium ions were washed out and potential BaSO$_4$ precipitates were redissolved, all samples

were flushed by a 0.1 mol L$^{-1}$ KCl solution over a period of five weeks and 150 mL KCl solution per sample. After the EC of the supernatant had stabilized at the EC of the KCl solution (tolerance maximally 2.5 %), the KCl rinsing process was stopped and the final 3-D images where taken of all samples. When the final images were taken the majority of the CAS was assumed to be saturated with Ba$^{2+}$. In the following we refer to these images as "Ba$^{2+}$ saturated" images. The average difference in the gray values of the soil solutions in the macropores between the reference images and the Ba$^{2+}$ saturated

images were within a range of 87 gray values thus, the soil solutions were assumed to have the same densities.

Three samples (SNO4, SNO6 and SNO10) were excluded from further analyses. SNO6 and SNO10 had been destroyed after the saturation process in the desiccator by a too rapid rise in air pressure. SNO4 had a very small hydraulic conductivity and could not be completed in this experiment.

## 2.5 Image processing

The software ImageJ/FIJI was used for image processing (Schindelin et al., 2012, 2015). The resolution of all 3-D images was reduced by a factor of 4 in order to decrease the image-processing time for the subsequent steps. Thus, the analyzed images had a resolution of 80 μm. The first step of the image analysis was to correct the image illumination to ensure that all



objects of the same density within individual images and across all images exhibited the same gray values. The mean gray values obtained for the aluminum wall and the KCl solution in the contrast image (Fig. 1) were used as the target gray values for the gray-scale standardization (21,418 and 16,225 respectively). All other gray values were scaled accordingly by the linear relationship between the target gray values and the initial gray values of the corresponding image according to Koestel

and Larsbo (2014). A 3-D unsharp mask with one pixel radius was applied to all images in order to increase sharpness. Both, the reference image and the $Ba^{2+}$ saturated image were binarized using a threshold gray value of 16,990, which was obtained from the joint histogram of all images that were to be binarized (Fig. 4). This was done to obtain the pore space for both the reference image and the corresponding $Ba^{2+}$ saturated image.

**2.6 Creating the difference images**

The images where registered using the ImageJ plugin "descriptor-based series registration (2-D/3-D)" by Preibisch et al. (2010) with the transformation model affine to account for minor soil deformations during the sample treatment. Thereafter, the difference images were obtained by subtracting the 3-D $Ba^{2+}$ saturated images from the reference 3-D images.

**2.7 Relationship between gray values and barium mass**

In order to estimate the $Ba^{2+}$ mass in the difference images as a proxy for the $CEC_{Ba}^{2+}$ the 3-D contrast image of $BaCl_2$, KCl

solutions and the aluminum wall was used as a reference (Fig. 1). The mean gray values of the two solutions were subtracted in order to obtain the maximal contrast in gray values ($\gamma_{SAT}$ = 2637) corresponding to the density contrast between the KCl and $BaCl_2$ solutions. The $BaCl_2$ mass ($m_{j,d}$ in mg) was then calculated according to Koestel and Larsbo (2014) using the Eq. (1).

$$m_{j,d} = \frac{\left( N_{Ba} - N_K \right)}{N_{Ba}} \frac{V_{VOX} C_{MAX}}{\gamma_{SAT}} \gamma_{j,d} \qquad \text{Eq. 1}$$

where $j$ is the voxel, $N_{Ba}$ = 56 and $N_K$ = 18 are the atomic numbers of barium and potassium, respectively, $V_{VOX}$ represents the voxels' volume ($5.12 * 10^{-7}$ cm³), $C_{MAX}$ the maximal possible increase in tracer solution (41.199 mg cm⁻³) and $\gamma$ the corresponding gray value. In Eq. 1 we assume that the CAS were predominantly occupied by $K^+$ when the reference images were taken and with $Ba^{2+}$ when the $Ba^{2+}$ saturated images were taken. In order to calculate an estimate of the $CEC_{Ba}^{2+}$ in cmol (+) kg⁻¹ soil the sum of the positive charged sites as occupied by $Ba^{2+}$ was calculated and related to the samples volume and

bulk density.

**2.8 Spatial distribution of cation exchange capacity**

In order to test whether the imaged CECs are elevated in macropore sheaths (400 μm distance from pore surface) as compared to the CECs in the soil matrix, the binarized 3-D images of the pore space for the reference and the $Ba^{2+}$ saturated





images were combined using the Image Calculator. This was done to account for any changes in pore space due to disturbances during the saturation periods. The resulting images of the combined pore space were dilated five times using the ImageJ plugin Process and its' function dilate 3-D. Subsequently, this dilated binary image was subtracted from the corresponding difference Image in a way that the resulting image showed the gray values from the difference image only outside the pore space and the dilation area (only the soil matrix). After inverting the dilated binary image it was subtracted from the difference image and resulting in a second image that represents only the gray values in the dilation area, thus the soil around the macropores. These images were used to assess the difference of the imaged density of CAS between the soil close to macropores and the soil in the matrix of all natural soil samples.

## 2.9 Spatial distribution of cation exchange capacity

For statistical analyses the open source software R (v0.98.1x) and RStudio (v3.2.5) was used (R Core Team, 2016). The relation between measured $CEC_{NH4}^{+}$ and the $CEC_{Ba}^{2+}$ obtained through image analyses was analyzed by a linear model. The $CEC_{Ba}^{2+}$ distribution around the macropores of the natural soil cores was graphically compared. For 3-D visualization the open-source software drishti was used (Limaye, 2012).

## 3 Results and Discussion

### 3.1 Image artifacts

Image artifacts in the difference images can originate due to soil movement after the reference image and before the $Ba^{2+}$ saturated image were taken. These will be visible as bright areas, if high gray values in the $Ba^{2+}$ saturated image (e.g. soil matrix) are subtracted from low gray values in the reference image (e.g. soil pore) and in dark areas, if the reverse is the case. The areas in the magnified difference images of SNO2 and SNO3 (marked with an X or a circle in Fig. 5) are due to such shifts. For SNO2 a shift only occurred locally, but for SNO3 this shift was visible throughout the soil column.

Swelling and shrinking of the soil in between the two scanning occasions could lead to similar image artifacts. In this case the most obvious artifacts would present themselves as brighter areas around macropores if the sample was swelling or darker areas if it was shrinking. However, the occurrence of artifacts due to swelling or shrinking was ruled out after scrutinizing the size and shape of respective macropores and it was found that none of them had changed (see Fig. 6 as an example).

In the difference image of SNO1 we found some brighter spots within the soil matrix that correspond to high density areas in the reference image (Fig. 6, circles). We hypothesize that these are porous iron or manganese oxides that either bind $Ba^{2+}$ and therefore visible as bright spots in the difference image, or these spots represent imaging artifacts resulting from the grayscale standardization. Since the applied grayscale standardization is only valid for imaged densities in between the density of the KCl solution and the density of the aluminum wall. In case the relationship between image gray-value and



material density is non-linear or changing with the introduction of a denser material (here $Ba^{2+}$), the resulting difference images may not represent the $Ba^{2+}$ densities only. This would especially be the case when estimating imaged densities larger than the aluminum wall (e.g. iron or manganese oxides).

## 3.2 Spatial heterogeneity of cation adsorption sites

Figure 7 shows cuboids extracted from the center of the reference images and the $Ba^{2+}$ saturated images from regions of interest with a quadratic horizontal cross-section (edge-length 14.4 mm) of SNO1 (height 17.76 mm), SNO3 (height 14.8 mm), SNO7 (height 20.32 mm) and SNO8 (height 20 mm). The $Ba^{2+}$ distribution is visualized by translucent areas (no or little $Ba^{2+}$), green (medium $Ba^{2+}$ density) and blue (high $Ba^{2+}$ density). Note that the color-scales in Fig. 7 are optimized for depicting the 3-D structures and therefore only semi-quantitative. The samples from the agricultural silty clay soil (SNO1

and SNO3) show a relatively uniform distribution of adsorbed $Ba^{2+}$ with some spots of high values within the matrix and some elevated values around macropores in SNO1. The heavy clay soil (SNO7) shows a high spatial heterogeneity in $Ba^{2+}$ densities with large areas of high values and large areas of very low values. The loamy sand soil (SON8) shows plenty of areas with no or little $Ba^{2+}$ adsorbed and some with locally distinct higher values. Figure 8 shows cross sections of the reference images for the seven intact soil samples and Fig. 9 shows the corresponding difference images, here the gray value

scale is quantitative. In Fig. 9 it is easy to differentiate between pores and soil matrix. Thus, most of the soils possess CAS abundant enough to be visualized by this method. The adsorbed $Ba^{2+}$ on CAS and its horizontal variances in the cross sections were particularly large in the heavy clay soils SNO5 and SNO7. In contrast, the other four undisturbed samples (SNO1, SNO2, SNO3 and SNO8) have lower contrasts and less $Ba^{2+}$ adsorbed. This pronounced difference between the samples reflects the variation in $CEC_{NH4}^{+}$ and OM contents between them (Table 1 and Table 2). The artificial sample

(SNO9) shows the highest gray values for OM aggregates, which is easy to differentiate from the clay aggregates and the sand. The difference in gray values of the clay aggregates and the surrounding sand is less distinct than the gray value difference of the OM aggregate and the sand. This reflects the variation of the respective $CEC_{NH4}^{+}$ (Table 2). In Fig. 10 the arithmetic means and standard deviations of the gray values in each horizontal cross section is plotted against depth for all soil samples. This confirms the observations made in Fig. 9 that SNO5 and SNO7 show the highest density of $Ba^{2+}$ adsorbed

to CAS and SNO7 the highest heterogeneity in a vertical profile. All the soils from the agricultural field-trial and the forest soil show lower $Ba^{2+}$ mass adsorbed and little vertical heterogeneity.

## 3.3 Cation adsorption sites and macropore space

The sample SNO1 shows brighter gray values around biopores. A magnification of these brighter areas shows that they surround the entire pore walls (with an approximate thickness of 0.4 mm, Fig. 6). This is likely due to a locally higher

$CEC_{Ba}^{2+}$ caused by accumulation of organic matter. Figure 6 shows a biopore in SNO1 that was created by an earthworm, judging on its shape and size. The $Ba^{2+}$ mass distribution calculated from the extracted gray values of the 3-D matrix space



and the 3-D space in the macropore sheaths of the difference images are presented in Fig. 11. For SNO3, SNO7 and SNO8 the median $Ba^{2+}$ masses within the matrix are lower than those within the sheaths. SNO2 and SNO5 show the opposite trend and no difference between the $Ba^{2+}$ masses of matrix and sheath is observed for SNO1. Hence, the observed increased $Ba^{2+}$ mass around the biopores depicted in Fig. 6 (middle) was specific to this macropore but was not observed for other

macropores in this sample. In general, there is little difference between the $Ba^{2+}$ mass distributions of the individual samples from the agricultural field-trial. This may be explained by the fact that the samples were taken from the plow layer in autumn after the field had been plowed and biopores were very rare or freshly formed and not yet coated with sufficient amounts of OM. With the exception of SNO5 the soils from the unplowed sites (marshy depression and forest soil) show higher $Ba^{2+}$ masses adsorbed within the macropore sheaths compared to the matrix. SNO7 originate from a heavy clay soil, where

biopores formed by faunal activity or root growth are not destroyed by plowing and therefore, the OM within them is redistributed to a lesser extent. SNO8 was sampled in a loamy sand soil, here the difference between the $Ba^{2+}$ mass distributions is likely due to the fact that the sand grains are surrounded by OM and fine material and therefore, more likely located within the matrix. In contrast, the macropore sheaths tend to contain higher amounts of OM and fine material. Sand grains typically exhibit a low CEC and seen as darker objects in the difference image compared to OM and fine material that

usually possess a higher CEC and are able to adsorb more $Ba^{2+}$. This can be seen in Fig. 11, where the median $Ba^{2+}$ mass of the matrix is below zero and its third quartile is exceeded by the median $Ba^{2+}$ mass of the macropore sheaths. $Ba^{2+}$ mass values below zero can be explained by a very low $Ba^{2+}$ mass in general and possibly in combination with small shifts in soil structure. In addition, the $Ba^{2+}$ contrast will be underestimated if the KCl treatment did not lead to a complete exchange of cations that are heavier than $K^+$ and were exchanged by $Ba^{2+}$ later.

**3.4 Comparison with ammonium acetate method**

The correlation of the $CEC_{NH4^+}$ measured in the laboratory with the $CEC_{Ba^{2+}}$ obtained from the difference images shows a significant relationship ($R^2 = 0.87$; p <0.01; Fig. 12). It confirms that the applied method captured the trend in CEC across the soil samples and strengthens the validity of our results obtained by the difference image analysis. The $CEC_{Ba^{2+}}$ underestimated the low CEC levels by the silty clay soils from the agricultural field and the loamy sand soil under forest

(SNO1, SNO2, SNO3 and SNO8) and overestimated the heavy clay soil that is rich in OM (SNO5 and SNO7). An underestimation of the CEC by our difference image analysis can result from multiple factors. The laboratory analysis of $CEC_{NH4^+}$ is based on experiments with sieved, finely grinded soils, whereas our results of $CEC_{Ba^{2+}}$ are based on undisturbed soils. Some soil surfaces in pores not accessible to the $BaCl_2$ solution in the undisturbed samples may not have been saturated with $Ba^{2+}$ and did not contribute to then $CEC_{Ba^{2+}}$. When investigated as sieved soils these surfaces get exposed and

more easily accessible, thus participating in ion exchange. Our method may estimate the physically accessible CAS more accurate than the $CEC_{NH4^+}$ method and therefore underestimates the CEC of the dense clay samples compared to the $CEC_{NH4^+}$ method. Using the two different index cations ($Ba^{2+}$ and $NH_4^+$) to estimate the CEC can also lead to different results





(Ciesielski et al., 1997; Jaremko and Kalembasa, 2014). In the presence of 2:1 clay minerals like smectites, $NH_4^+$ may be trapped in the interfoliaceous cavities (Essington, 2004a; Pansu and Gautheyrou, 2006). Furthermore, competition and displacement of $Ba^{2+}$ on clay mineral surface adsorption sites by $K^+$ can be rather strong due to their similar ionic radii (Kabata-Pendias, 2010). Therefore, the KCl rinsing process may have i) a reduced $Ba^{2+}$ densities locally as well as the X-ray
photon attenuation on clay mineral surfaces and ii) underestimated $CEC_{Ba^{2+}}$ of clay-rich samples in contrast to organic matter rich samples where $Ba^{2+}$ can be bound in OM complexes (Bodek et al., 1988; Bradl, 2004; Lee et al., 2007; Pichtel et al., 2000). The low $CEC_{Ba^{2+}}$ values of SNO8 seem to contradict this hypothesis. However, SNO8 originated from a loamy sand soil under a pine forest, and its soil OM appeared little humified, therefore the density of functional groups and the potential to develop a larger CEC may not have been fully reached (Essington, 2004b). Additionally, the low pH of this sample may
have led to an underestimation of the $CEC_{Ba^{2+}}$ compared to the buffered $CEC_{NH4^+}$ (Skinner et al., 2001).

## 4 Conclusions

We have shown that a modern industrial X-ray scanner is capable of providing 3-D images that can be used to map the cation adsorption sites (CAS) in undisturbed soil cores by difference image analyses. Furthermore, $Ba^{2+}$ provides enough contrast to assess the 3-D distribution of CAS in soil. All undisturbed soil samples showed some degree of a spatially heterogeneous
distribution of CAS most of them down to a millimeter scale. However, no clear relationship between the location of CAS and the macropore structure was found. Even though our method deviated from the common ammonium acetate method to some extent, the results correlated significantly. This deviation may be due to several factors. The most likely are a lower accessibility of CAS in the undisturbed soils used here compared to the ammonium acetate method that is based on sieved soils. A competition between $K^+$ and $Ba^{2+}$ on the ion exchange sites on clay surfaces that may have led to a reduction in local
$Ba^{2+}$ densities in the clay rich samples. To further assess the discrepancy between these two methods a larger set of soil columns could provide detailed information on systematic deviations due to physical or chemical soil properties. It could also provide us with further valuable information on the accessibility of CAS and aid the development of more realistic soil system model. The difference image quality could be improved by using the absorption edge technique and a monochromatic X-ray beam as is available on synchrotron facilities. This would eliminate the negative effects on the difference image
quality by soil aggregate displacement or a possible change in the relationship between the gray values and object densities, although it would reduce the sample size.

## 5 Competing interests

The authors declare that they have no conflict of interest.

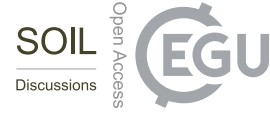

# 6 Acknowledgment

We are grateful to Nicholas Jarvis for helpful discussions and language advice.





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



**Table 1.** Location of each sampling site, Organic carbon (Org. C), clay, silt and sand contents.

| Site | Location | Org. C [%] | Clay [%] | Silt [%] | Sand [%] |
|---|---|---|---|---|---|
| Kungsängen (3-5 cm depth) | 59.836361, 17.6875 | 2.17 | 55.5 | 41 | 3.6 |
| Kungsängen (35-38 cm depth) | 59.836361, 17.6875 | 2.34 | 53.9 | 42.4 | 3.7 |
| Periodically water logged soil | 59.823972, 17.6623 | 4.78 | 79.9 | 20 | 0.3 |
| Forest soil | 59.824194, 17.6633 | 4.76 | 8.9 | 10.7 | 80.3 |



**Table 2.** Soil texture classes, sampling site, pH, cation exchange capacity ($CEC_{NH4}^{+}$) and bulk density (BD) for each sample number (SNO).

| SNO | Soil texture | Site | pH [$H_2O$] | CEC [cmol kg$^{-1}$] | BD [g cm$^{-3}$] |
|---|---|---|---|---|---|
| 1 | Silty clay | Kungsängen (3-5 cm depth) | 6.5 | 20.3 | 1.14 |
| 2 | Silty clay | Kungsängen (35-38 cm depth) | 6.5 | 19.1 | 1.30 |
| 3 | Silty clay | Kungsängen (3-5 cm depth) | 6.3 | 19.9 | 0.87 |
| 4 | Silty clay | Kungsängen (35-38 cm depth) | 6.2 | 20.4 | 1.20 |
| 5 | Heavy clay | Periodically water logged soil | 7.1 | 47.9 | 0.48 |
| 6 | Heavy clay | Periodically water logged soil | 6.8 | 22.0 | 0.41 |
| 7 | Heavy clay | Periodically water logged soil | 6.8 | 22.0 | 0.56 |
| 8 | Loamy sand | Forest soil | 5.1 | 13.1 | 0.96 |
| 9 & 10 | Fine sand | | 4.9 | 3.0 | |
| 9 & 10 | Clay aggregates | | 7.0 | 16.9 | |
| 9 & 10 | Peat aggregates | | 6.7 | 45.7 | |



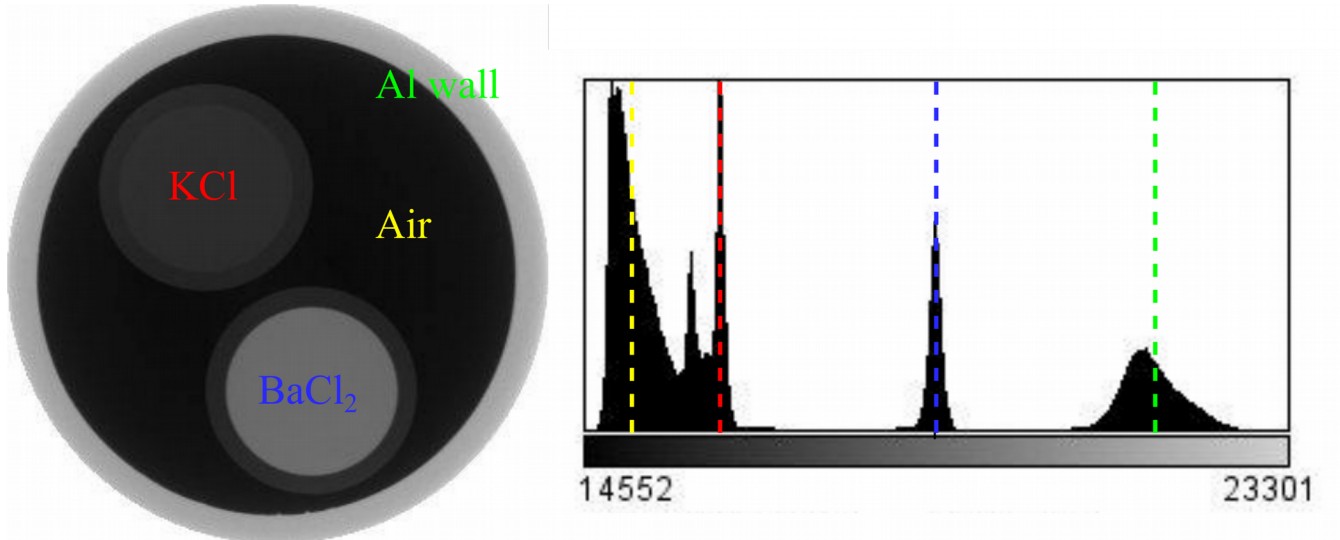

**Figure 1:** Contrast image of the aluminum cylinder wall, KCl and $BaCl_2$ solution and the air (left). The corresponding histogram on the right with the mean values for the aluminum cylinder wall (green), KCl (red) and $BaCl_2$ (blue) solution and the air (yellow) indicated as dotted lines.




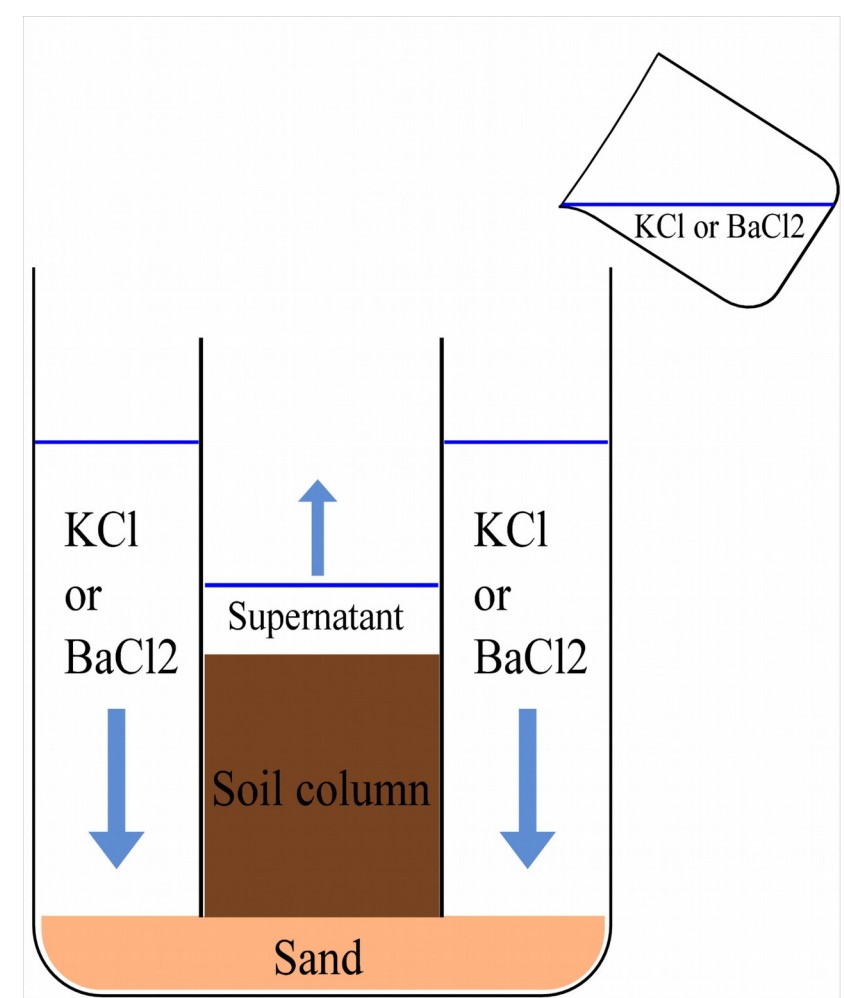

**Figure 2:** Set-up for the KCl or BaCl2 saturation process.



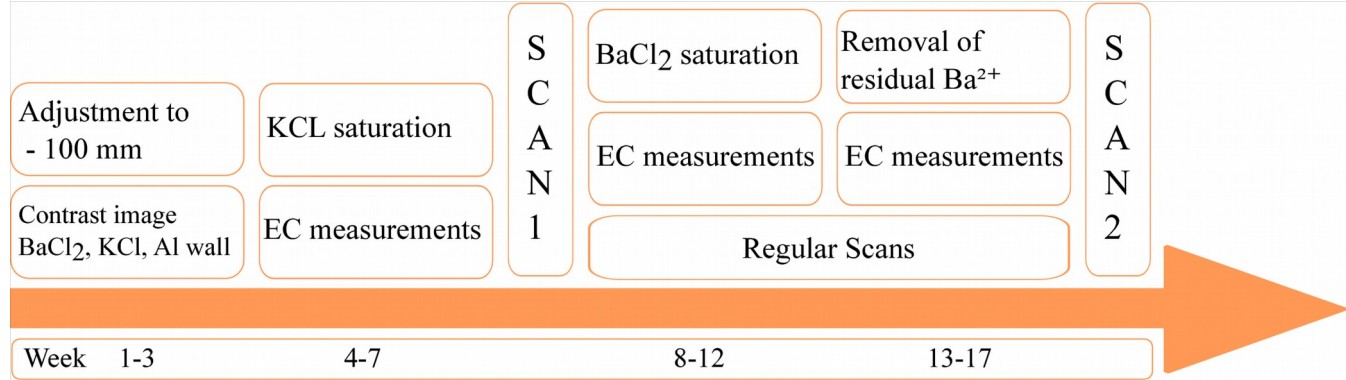

**Figure 3:** Experimental time schedule. X-ray scans were performed after KCl treatment to obtain the reference image (Scan 1) and after removal of residual $Ba^{2+}$ ions to obtain the $Ba^{2+}$ saturated image (Scan 2).

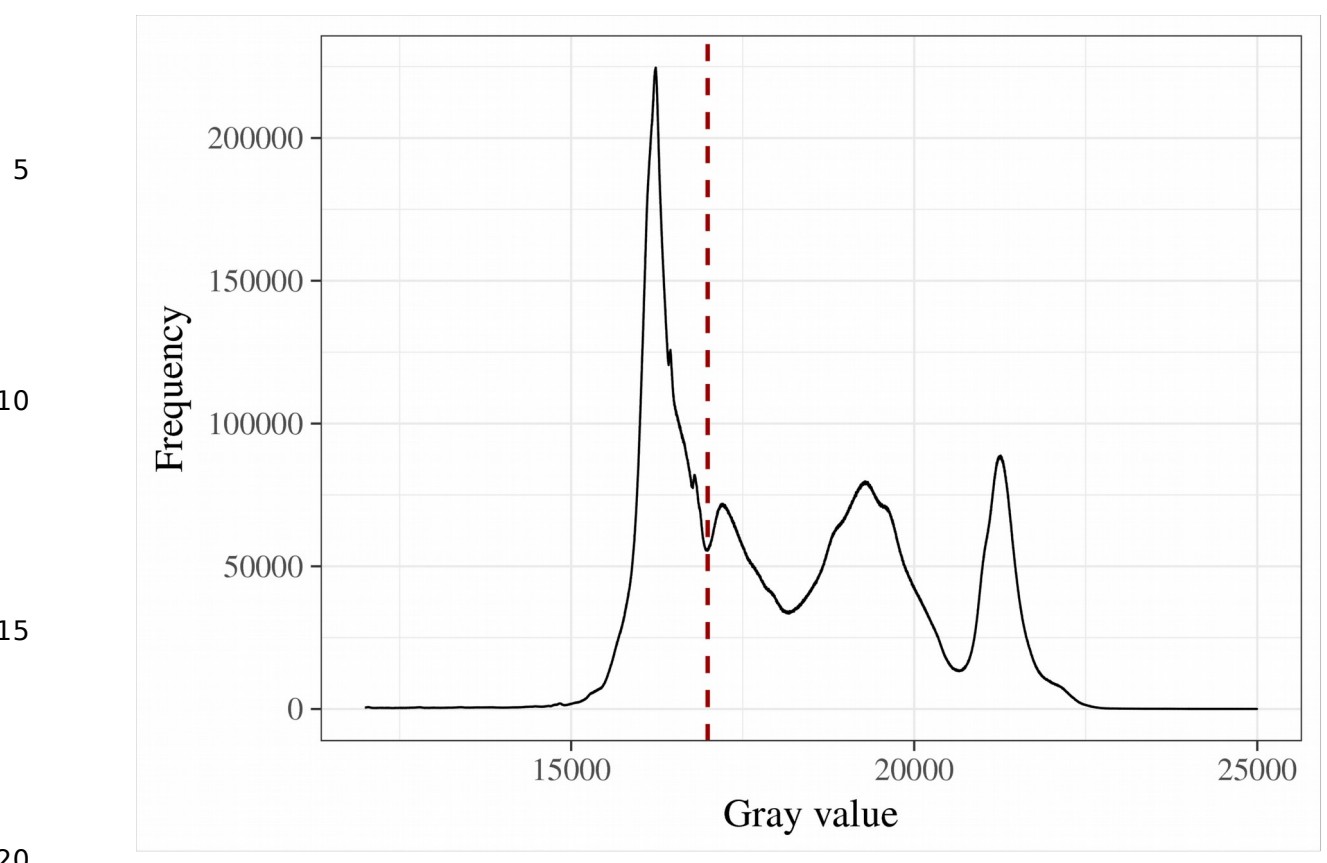

**Figure 4:** The joint histogram of the reference and $Ba^{2+}$ saturated images. The red dotted line represents the segmentation threshold that was used for binarization.



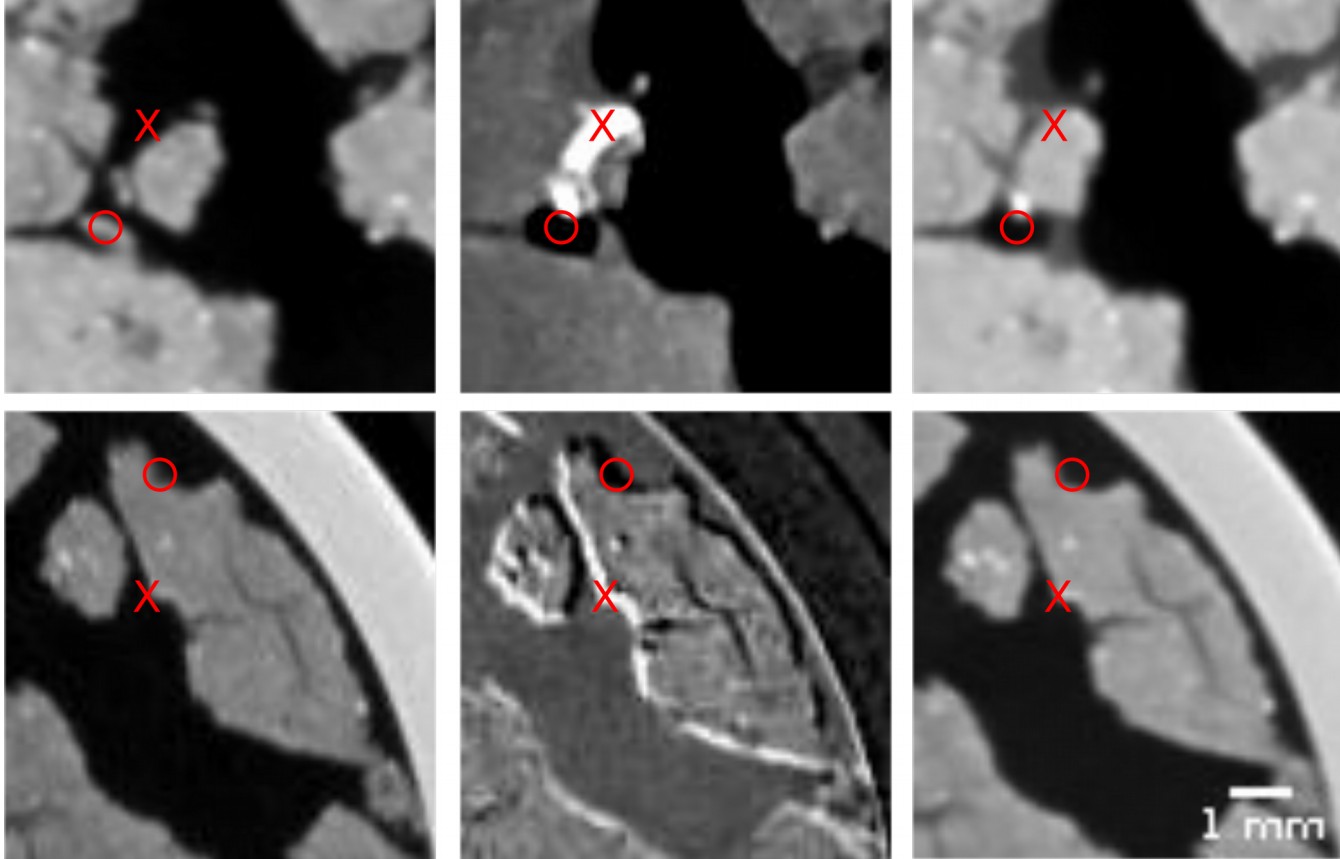

**Figure 5:** Effect of aggregates movement on the difference image of SNO2 (top) and SNO3 (bottom). Reference image (left), difference image (middle), Ba²⁺ saturated image (right). The red cross and circle indicate the identical coordinates in all three images. The movement of one soil aggregate resulted in very high gray values (X) or very low gray values (circle) in the difference image.



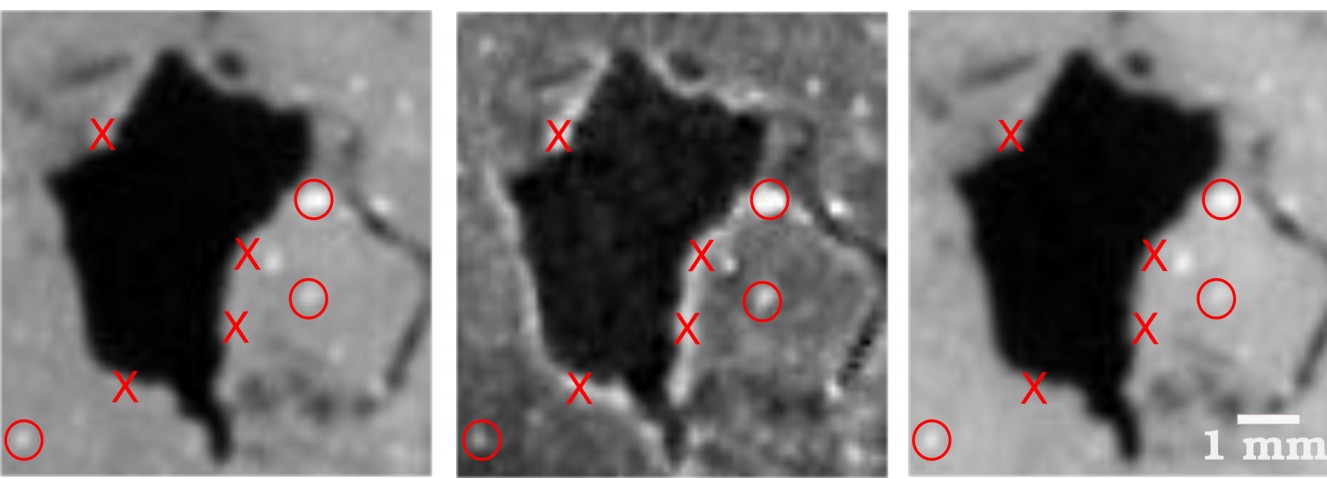

**Figure 6:** Magnification of a macropore from SNO1. Reference image (left), difference image (middle), $Ba^{2+}$ saturated image (right). The X and the circle indicate the identical coordinates in the images.



**Figure 7:** Cuboids with horizontal edge-lengths of 14.4 mm extracted from the center of SNO1 (height 17.76 mm), SNO3 (height 14.8 mm), SNO7 (height 20.32 mm) and SNO8 (height 20 mm). The reference images (left) and the respective $Ba^{2+}$ distributions (right) are shown. Translucent and dark gray values depict low density and bright gray-values depict high density regions. The $Ba^{2+}$ distribution is visualized by translucent (no or little $Ba^{2+}$), green (medium $Ba^{2+}$ density) and blue (high $Ba^{2+}$ density) colors. Note that the color-scales in are optimized for depicting the 3-D structures. They are therefore only semi-quantitative.



**Figure 8:** Cross sections of the reference images of the seven intact soil samples. Depth from the soil surface: SNO1, SNO2 and SNO3 at SNO8 mm, SNO7 and 8 at 9 mm, SNO5 at 15 mm and SNO9 at 5 mm.



**Figure 9:** Difference images of the seven intact soil samples. The gray scale represents the Ba$^{2+}$ mass (M$_{Ba}$) in µg per voxel (vx). Depth from the soil surface: SNO1, SNO2 and SNO3 at SNO8 mm, SNO7 and SNO8 at 9 mm, SNO5 at 15 mm and SNO9 at 5 mm.





**Figure 10:** Vertical profiles of the Ba$^{2+}$ mass distributions of the seven natural soil samples. Arithmetic mean (left) and standard deviation (right).







**Figure 11:** Comparison of the $Ba^{2+}$ mass distributions within the matrix (red) and within the macropore sheaths (blue) for all natural soil samples.





**Figure 12:** Relation between the CEC (cmol(+) kg$^{-1}$) measured with NH$_4^+$ and the CEC$_{Ba}^{2+}$ obtained from the difference image analysis of the natural soil samples, blue line represents a linear model with its 95 % confidence interval in gray (p < 0.01). Dotted line has slope 1.