# Peer review of "Quantitative imaging of the 3-D distribution of cation adsorption sites in undisturbed soil"

_SOIL, 2017_

## Referee Comment (RC1) · Anonymous Referee #1 · 28 Jun 2017

Review Comments on
Keck et al.; Quantitative imaging of 3-D distribution of cation adsorption sites in undisturbed soil.

This manuscript used X-ray tomographic contrast imaging to determine the spatial heterogeneity of cation adsorption sites (CAS) with 80-micron spatial resolution. The authors showed remarkable consistency of barium distributions in samples from a silty agricultural soil compared with samples from more organic-matter rich, more clayey forest and bog soils.

Specific comments

1. p. 5, L. 19-23 and p. 6, L. 22-23: How much of the $Ba^{2+}$ is likely removed from the CAS with a 0.1 M KCl wash?  Although there should be a high proportion of divalent to monovalent cation in the exchanger phase for an equal ratio of aqueous cations, it seems that a 0.1 M KCl wash would remove a significant portion of $Ba^{2+}$ from the CAS? Perhaps the assumption that all CAS was saturated with $Ba^{2+}$ after the 0.1 M KCl wash (p. 6, L. 22-23) is not completely valid.  An alternative is to call the $Ba^{2+}$-saturated sites "sites of higher affinity for $Ba^{2+}$").  Actually, the authors address this possibility in their discussion of CEC relationships (p. 10), but perhaps the possibility could be raised in the methods section also.

2. Fig. 5. Is there a "cutoff" difference value to assess what regions of the X-ray images are artifacts? For example, it is not clear in Fig. 5 how it was deduced that the SNO3 sample had a global shift, whereas the "very bright" or "very dark" areas in the SNO2 image was apparent. Also, can a brightness scale bar be put on Fig. 5 (and 6) to give an idea of the difference scale of the various grey shades?

3. It would be helpful to have more details in the methods on how the artificial sample (SNO9) was prepared.

4. Because the manuscript primarily discusses a new X-ray imaging technique, the conclusions could be expanded to present opportunities for using this technique other than for CAS mapping.  For example, I found the discussion of organic-lined biopores to be an interesting observation. Could, for example, the technique presented here be used to study such pores in more detail, e.g., at the original spatial resolution of the data for smaller sample volumes?

---

## Referee Comment (RC2) · Anonymous Referee #2 · 14 Jul 2017

This is a very interesting study determining the spatial distribution of cation adsorption sites in undisturbed soil columns. Until now, our knowledge about the spatial heterogeneity down to the millimetre scale (and smaller) of this important soil property is very limited and quantitative approaches are strongly needed. The authors compared a 3-D quantitative imaging approach to determine the cation exchange capacity with a conventional method. In general the manuscript is of high quality and well written. I have just a few comments related to the methodological approach including one suggestion for a preliminary test (small amount of work).

Introduction: The list of references on current X-ray CT studies (page 2 line 25 to page 3 line 18) is limited on what seems to be a subjective selection by the authors, but that cannot be hold against them as the field is growing quickly and it is neither possible nor

desirable to reach a comprehensive overview here. The references are all up to date!

Methods: Page 4, line 8: Please add a reference for the ammonium acetate method you used. Page 4, line 14: You referred to the "loss of ignition method" for the analyses of carbon in your soils but you used a CNS analyser. Please check / clarify. page 4 line 23: "inverted" should be replaced by "reconstructed into 3-D tomograms"

Reference gray values for Al, K and Ba in the contrast image (Fig. 1): What was the motivation to use different concentration for the two solutes (0.1M KCl vs 0.3M BaCl2)? Why did you keep the remaining volume inside the soil column air-filled? In a more realistic scenario it should be filled with dry soil at the same or at least the average bulk density of all investigated samples to mimic the X-ray attenuation by the soil matrix. For instance, the reference gray value for KCl will be below 16225 if pure KCl was detected inside soil, because the photon flux will already be attenuated during the passage of the soil matrix, and it is difficult to estimate by how much. I would strongly advice to do such a preliminary test with the same solution columns surrounded by differently packed soil that covers the range of bulk densities reported in Table 2, analyse the effect on the reference gray values of the different materials and add this information as supplementary material. The amount of work to do this is small. If the changes turn out to be small, then you can use this as an asset of your approach to use polychromatic X-rays to measure cation adsorption sites. As this is really the first study in this respect it should lay out the foundations as thoroughly as possible. Additional information about the interplay between attenuation of polychromatic X-rays in water and the soil matrix can be retrieved e.g. in Weller et al. (2017).

Page 6 line 11: To my knowledge an affine transformation cannot account for local deformations, but only for a change in position and perhaps global distortion of the sample.

Page 6, relationship between gray values and barium mass: It is not clear to me, also after consulting Koestel & Larsbo (2014), how Cmax is determined. Is it the hypothetical mass of Ba in one voxel assuming 0.3M BaCl is reached in a pure pore voxel, i.e. no partial filling of that voxel with the solid phase? Also, readers might wonder how changes in background porosity might influence the interpretation of the estimated BaCl2 mass. Does the same increase in m (BaCl2 mass) always result in the same increase in gamma (gray value), no matter whether a voxel is partially filled with pores by, say, 30% or 70%?

Page 6-7, Spatial distribution of cation exchange capacity: In order to estimate CEC from m, you need to know the mass of soil in the two regions of interest (soil matrix vs. macropore walls). Do you estimate the (fluctuating) mass of soil per voxel and cumulate this over all voxels in the respective regions?

Table 1: Please indicate sampling depths and soil horizons for all soils you used

References: Weller, U., F. Leuther, S. Schlüter, H.-J. Vogel: Quantitative analysis of water infiltration in soil cores using x-ray, Vadose Zone Journal (in press), url: https://dl.sciencesocieties.org/publications/vzj/first-look

---

## Author Comment (AC1) · 10 Aug 2017

**Reply to the Comments of Reviewer 1.**

Keck et al.; Quantitative imaging of the 3-D distribution of cation adsorption sites in undisturbed soil.

Thank you for reading our manuscript carefully and for the time and effort you spend to comment on it.

**Comment 1:**
p. 5, L. 19--23 and p. 6, L. 22--23: How much of the Ba2+ is likely removed from the CAS with a 0.1 M KCl wash? Although there should be a high proportion of divalent to monovalent cation in the exchanger phase for an equal ratio of aqueous cations, it seems that a 0.1 M KCl wash would remove a significant portion of Ba2+ from the CAS? Perhaps the assumption that all CAS was saturated with Ba2+ after the 0.1 M KCl wash (p. 6, L. 22-23) is not completely valid. An alternative is to call the Ba2+--saturated sites "sites of higher affinity for Ba2+"). Actually, the authors address this possibility in their discussion of CEC relationships (p. 10), but perhaps the possibility could be raised in the methods section also.

**Reply to comment 1:**
Thank you for this comment. We agree that not all CAS may be completely saturated with Ba2+ after the KCl wash. We also think it is a good idea to address this issue not only in the discussion but also in the material and method section of the revised manuscript and will adapt the term 'Ba2+ saturated sites' accordingly.

**Comment 2:**
Fig. 5. Is there a "cutoff" difference value to assess what regions of the X--ray images are artifacts? For example, it is not clear in Fig. 5 how it was deduced that the SNO3 sample had a global shift whereas the "very bright" or "very dark" areas in the SNO2 image was apparent. Also, can a brightness scale bar be put on Fig. 5 (and 6) to give an idea of the difference scale of the various grey shades?

**Reply to comment 2:**
In Fig. 1 we visualised the global gray value (GV) distribution of the soil columns of samples number (SNO) 1, 2 and 3. All difference images show histograms with a small peak around GV -1300 (black vertical line). These correspond to air bubbles that formed after the reference images were taken. A difference between SNO1 with very few artefacts and SNO2 and 3 with more abundant artefacts is also apparent. The histograms of SNO2 and SNO3 have two plateaus in gray value abundance at approx. GV +/-1500 to +/-4500 and GV +/-1500 to +/-3750 respectively (red horizontal lines in Fig. 1). These plateaus represent local particle shifts within the samples that occurred after the reference images were taken (for an example see the highlighted areas in Fig. 2).
This can be assumed following the reasoning that a particle shift will lead to bright areas if high GV are subtracted from low GV and in dark areas, if the reverse is the case. This means that per shifted aggregate there is usually one side that is 'framed' by higher GV and one side that is 'framed' by lower GV. Both in equal proportions. This may be a more objective way to characterise artefacts due to shifts in difference images. Determining and integrating the plateaus could be used to quantify registration errors in future studies. A cut-off value can be determined at the beginning of the plateaus (approximately at a GV +/- 1500), however this might exclude some GV originating from regions of enhanced barium adsorption.
We have included a gray scale bar for Fig. 5 and 6 (see below, Fig. 2 and 3).

**Comment 3:**
It would be helpful to have more details in the methods on how the artificial sample (SNO9) was prepared.

**Reply to comment 3:**
Thank you, we will include more details on the preparation of the artificial sample.

**Comment 4:**
Because the manuscript primarily discusses a new X--ray imaging technique, the conclusions could be expanded to present opportunities for using this technique other than for CAS mapping. For example, I found the discussion of organic--lined biopores to be an interesting observation. Could, for example, the technique presented here be used to study such pores in more detail, e.g., at the original spatial resolution of the data for smaller sample volumes?

**Reply to comment 4:**
Yes, we believe that it is possible to use this method for mapping organic matter within undisturbed soil cores, especially when it comes to the organic-lined biopores. For this purpose the KCl rinsing process should be somewhat longer and one could consider to increase the KCL concentration. This would make it more likely that most of the $Ba^{2+}$ bound to clay surfaces and other exchange sites is replaced by $K^+$, whereas the $B^{2+}$ bound in complexes to organic matter would stay in place.

Heavy anions could be used as contrast agents for imaging the soil organic matter instead of barium (e.g. $I^-$, $Br^-$, $WO_4^{2-}$ or $MoO_4^{2-}$). When used on soils from temperate climate regions these may have the advantage that the CEC is not biasing the results.

It is furthermore possible to improve the resolution. However, an improved resolution mainly depends on the sample size. The smaller the sample the better the resolution. Note that the maximal resolution also depends on the hardware used (X-ray scanner and computer) and its configuration. After some preliminary tests we found that the scanner used in this study (GE Phoenix v|tome|x m) is capable of taking images at a resolution down to 5 µm at a soil column with a diameter of 8 mm. Others have reported resolution down to 1 µm when using X-ray scanner optimized for smaller sample sizes (e.g. Tippkötter et al., 2009). By using a monochromatic X-ray source Voltolini et al. (2017) imaged soil micro-aggregates with a sub-micron resolution.

Tippkötter, R., Eickhorst, T., Taubner, H., Gredner, B., Rademaker, G., 2009. Detection of soil water in macropores of undisturbed soil using microfocus X-ray tube computerized tomography (μCT). Soil Tillage Res. 105, 12–20. doi:10.1016/j.still.2009.05.001

Voltolini, M., Taş, N., Wang, S., Brodie, E.L., Ajo-Franklin, J.B., 2017. Quantitative characterization of soil micro-aggregates: New opportunities from sub-micron resolution synchrotron X-ray microtomography. Geoderma 305, 382–393. doi:10.1016/j.geoderma.2017.06.005

**Figures:**

[Figure]

[Figure]

[Figure]

**Figure 1:** Global gray value distribution for SNO1, SNO2 and SNO3.

[Figure]

**Figure 2:** Effect of aggregates movement on the difference image of SNO2 (top) and SNO3 (bottom). Reference image (left), difference image (middle) and the image of Ba$^{2+}$ treated soil (right). The red cross and circle indicate the identical coordinates in all three images. The movement of one soil aggregate resulted in very high gray values (red marks) or very low gray values (blue marks) in the difference image. Note that the reference images and the image of Ba$^{2+}$ treated soil share the same gray value calibration bar.

[Figure]

**Figure 3:** Magnification of a macropore from SNO1. Reference image (left), difference image (middle) and the image of Ba$^{2+}$ treated soil (right) (right). The X and the circle indicate the identical coordinates in the images. Note that the reference image and image of Ba$^{2+}$ treated soil share the same gray value calibration bar.

---

## Author Comment (AC2) · 10 Aug 2017

**Reply to the Comments of Reviewer #2.**

Keck et al.; Quantitative imaging of the 3-D distribution of cation adsorption sites in undisturbed soil.

**Comment 1:**

This is a very interesting study determining the spatial distribution of cation adsorption sites in undisturbed soil columns. Until now, our knowledge about the spatial heterogeneity down to the millimetre scale (and smaller) of this important soil property is very limited and quantitative approaches are strongly needed. The authors compared a 3-D quantitative imaging approach to determine the cation exchange capacity with a conventional method. In general the manuscript is of high quality and well written. I have just a few comments related to the methodological approach including one suggestion for a preliminary test (small amount of work).

**Reply 1:**

Thank you for your interest in our manuscript and for the time and effort you spend to comment on it.

**Comment 2:**

Introduction: The list of references on current X-ray CT studies (page 2 line 25 to page 3 line 18) is limited on what seems to be a subjective selection by the authors, but that cannot be hold against them as the field is growing quickly and it is neither possible nor desirable to reach a comprehensive overview here. The references are all up to date!

**Reply 2:**

Thank you.

**Comment 3:**

Methods: Page 4, line 8: Please add a reference for the ammonium acetate method you used. Page 4, line 14: You referred to the "loss of ignition method" for the analyses of carbon in your soils but you used a CNS analyser. Please check / clarify. page 4 line 23: "inverted" should be replaced by "reconstructed into 3-D tomograms"

**Reply 3:**

We agree that it is a very valuable information to include and will refer the reader to the description of the ammonium acetate method by (Thomas, 1982). The loss of ignition analysis was done on a TruMac CN analyser from LECO, not as mentioned in the manuscript on a TruMac CNS analyser. We will also change 'inverted' (p.4 line 23) to 'reconstructed into 3-D tomograms'.

**Comment 4:**

Reference gray values for Al, K and Ba in the contrast image (Fig. 1): What was the motivation to use different concentration for the two solutes (0.1M KCl vs 0.3M BaCl2)? Why did you keep the remaining volume inside the soil column air-filled? In a more realistic scenario it should be filled with dry soil at the same or at least the average bulk density of all investigated samples to mimic the X-ray attenuation by the soil matrix. For instance, the reference gray value for KCl will be below

16225 if pure KCl was detected inside soil, because the photon flux will already be attenuated during the passage of the soil matrix, and it is difficult to estimate by how much. I would strongly advice to do such a preliminary test with the same solution columns surrounded by differently packed soil that covers the range of bulk densities reported in Table 2, analyse the effect on the reference gray values of the different materials and add this information as supplementary material. The amount of work to do this is small. If the changes turn out to be small, then you can use this as an asset of your approach to use polychromatic X-rays to measure cation adsorption sites. As this is really the first study in this respect it should lay out the foundations as thoroughly as possible.

and the soil matrix can be retrieved e.g. in Weller et al. (2017).

**Reply 4:**

Thank you for this very valuable advice!

Concerning your first question, we used the KCl solution to flush out the residual  $Ba^{2+}$  ions and chose to set the concentration lower that of the  $BaCl_2$  solution. A KCl solution, rather than deionised water was chosen to avoid structural changes of the clay minerals. These changes could be expected if the KCl solution was either too strong or too weak.

Your second question is indeed very interesting to elaborate on. We did some preliminary tests on the effects of higher bulk densities within the aluminium columns of the calibration images. The aluminium columns were filled with differently packed soil, otherwise the procedure was identical to the description in the manuscript. Please see the gray value distributions of the KCl and the BaCl2 solutions in Fig. 1 and Fig. 2 and corresponding cross-sections in Fig. 3. The average increase in contrast of the three samples with packed soil compared to the sample without soil (air-filled) was 9.99 % (Tab. 1 for more details). Considering that these packed soils exceeded the bulk densities of the samples described in our manuscript this increase in contrast is the maximum expectable increase. Therefore, it would mean that our estimates of the cation exchange capacity (CEC) are underestimated by a maximum of 10 %. Fig. 4 illustrates the implications of such an increase on our estimates of the CEC.

**Comment 5:**

Page 6 line 11: To my knowledge an affine transformation cannot account for local deformations, but only for a change in position and perhaps global distortion of the sample.

**Reply 5:**

Thank you, we will change our wording at the respective position.

**Comment 6:**

Page 6, relationship between gray values and barium mass: It is not clear to me, also after consulting Koestel and Larsbo (2014), how Cmax is determined. Is it the hypothetical mass of Ba in one voxel assuming 0.3M BaCl is reached in a pure pore voxel, i.e. no partial filling of that voxel with the solid phase? Also, readers might wonder how changes in background porosity might influence the interpretation of the estimated BaCl2 mass. Does the same increase in m (BaCl2 mass) always result in the same increase in gamma (gray value), no matter whether a voxel is partially filled with pores by, say, 30% or 70%?

**Reply 6:**

In eq. 1  $C_{max}$  refers to the maximum possible increase in Ba2+ concentration (0.3 m Ba2+ or 41.199 mg cm-3) i.e. a voxel filled with BaCl2 solution (no partial volume voxel).

We are assuming that the increase in  $Ba^{2+}$  mass is linearly related to GV in the difference images, irrespective of the voxel porosity. We are however aware that this assumption is only an approximation (see reply to comment 4). With reference to the results of the additional experiment conducted in connection with comment 4, we suspect that partial volume effects are of subordinate importance. We however agree that such possible effects should be investigated in future experiments.

**Comment 7:**

Page 6-7, Spatial distribution of cation exchange capacity: In order to estimate CEC from m, you need to know the mass of soil in the two regions of interest (soil matrix vs. macropore walls). Do you estimate the (fluctuating) mass of soil per voxel and cumulate this over all voxels in the respective regions?

**Reply 7:**

Thank you for this comment. Actually we did not estimate the CEC within the matrix or the macropore walls itself. Here we estimated the barium mass only. We have realised this is described somewhat misleading (page 6 line 27). We will change the wording from:

'In order to test whether the imaged CECs are elevated in macropore sheaths (400  $\mu$ m distance from pore surface) as compared to the CECs in the soil matrix [...]'

to:

'To test whether the imaged barium concentrations as proxies for the CECs are elevated in macropore sheaths (400  $\mu$ m distance from pore surface) as compared to the barium concentrations in the soil matrix [...]'

**Comment 8:**

Table 1: Please indicate sampling depths and soil horizons for all soils you used.

**Reply 8:**

We will add the information on sampling depth to the bog and forest soil in Table 1.

**References:**

- Koestel, J., Larsbo, M., 2014. Imaging and quantification of preferential solute transport in soil macropores. Water Resour. Res. 50, 4357–4378. doi:10.1002/2014WR015351
- Thomas, G.W., 1982. Exchangeable cations, in: Methods of Soil Analysis. Part 2. Chemical and Microbiological Properties. Madison, USA, pp. 154–157.
- Weller, U., F. Leuther, S. Schlüter, H.-J. Vogel: Quantitative analysis of water infiltration in soil cores using x-ray, Vadose Zone Journal (in press), url: https://dl.sciencesocieties.org/publications/vzj/first-look

**Table 1.** Average gray values (GV) of the KCl and  $BaCl_2$  solutions of contrast images with different bulk densities (BD) and its effect on the resulting contrast between the GV of the  $BaCl_2$  and the KCl solutions.

|                                                   | Air      | BD 1.12 g cm -3
Silty clay | BD 1.43 g cm -3
Silty clay | BD 1.66 g cm -3
Sand | units |
|---------------------------------------------------|----------|------------------------------------------|------------------------------------------|------------------------------------|-------|
| KCl solution                                      | 16254.30 | 16026.01                                 | 16184.68                                 | 15945.33                           | GV    |
| BaCl 2 solution                        | 19043.46 | 19063.68                                 | 19136.38                                 | 19159.73                           | GV    |
| Resulting contrast                                | 2789.16  | 3037.68                                  | 2951.70                                  | 3214.41                            | GV    |
| Difference in contrast compared to the air sample | 0        | 8.91                                     | 5.83                                     | 15.25                              | %     |

**Figure 1.** Gray value distributions of the KCl solution from four different contrast images. Yellow: with an air-filled aluminium column, blue, green and purple: with packed soil but different bulk densities (BD).